computational biology/bioinformatics/software

information theory, CoHSI, proteome, software, music

**Author for correspondence:**
Les Hatton
e-mail: lesh@oakcomp.co.uk

# Strong evidence of an information-theoretical conservation principle linking all discrete systems

## Les Hatton[1] and Gregory Warr[2]

[1]Faculty of Science, Engineering and Computing, Kingston University, London, UK
[2]Department of Biochemistry and Molecular Biology, Medical University of South Carolina, Charleston, SC, USA

LH, 0000-0003-2226-3453

Diverse discrete systems share common global properties that lack a unifying theoretical explanation. However, constraining the simplest measure of total information (Hartley–Shannon) in a statistical mechanics framework reveals a principle, the conservation of Hartley–Shannon information (CoHSI) that directly predicts both known and unsuspected common properties of discrete systems, as borne out in the diverse systems of computer software, proteins and music. Discrete systems fall into two categories distinguished by their structure: *heterogeneous* systems in which there is a distinguishable order of assembly of the system's components from an alphabet of unique tokens (e.g. proteins assembled from an alphabet of amino acids), and *homogeneous* systems in which unique tokens are simply binned, counted and rank ordered. Heterogeneous systems are characterized by an implicit distribution of component lengths, with sharp unimodal peak (containing the majority of components) and a power-law tail, whereas homogeneous systems reduce naturally to Zipf's Law but with a drooping tail in the distribution. We also confirm predictions that very long components are inevitable for heterogeneous systems; that discrete systems can exhibit simultaneously both heterogeneous and homogeneous behaviour; and that in systems with more than one consistent token alphabet (e.g. digital music), the alphabets themselves show a power-law relationship.

## 1. Introduction

Discrete systems, i.e. systems that comprise pieces that can be consistently counted, are everywhere in the inanimate world (e.g. matter itself), the biological world (e.g. DNA, proteins, species) and the world of human endeavour and creativity (e.g.

computer software, written texts, digital music). The possibility that the global properties of such discrete systems might be shaped by a single unifying principle is attractive but challenging given their diverse provenance and the clear knowledge that discrete systems have been shaped by completely different forces from the fundamental laws of physics and chemistry to evolution and human creativity. The properties of discrete systems have been examined from many perspectives (e.g. [1–13]), and although a number of explanations have been advanced to explain the behaviour of particular systems, no satisfactory unifying theory has emerged. A significant complicating factor is that the common property of discrete systems most frequently observed is the rank order/frequency power-law relationship exemplified by Zipf's Law [14] for written texts; when specific words in a text are counted and their frequencies rank ordered, the log–log plot of frequency versus rank is a straight line. Power-law relationships are ubiquitous [15] and occur in natural (including societal) phenomena as diverse as solar flares, earthquakes, power failures, distribution of wealth, wildfires and neural functions; attempts to explain all power laws in nature by a single underlying physical principle such as self-organized criticality are considered untenable [16], with the favoured explanation for the ubiquity of power-law relationships being that diverse processes can lead to the same outcome. Thus, in seeking to uncover a principle that guides the shared behaviour of discrete systems, we must engage with more than Zipf's Law [17].

A theory of discrete systems should satisfy, at a minimum, the following criteria: (i) it must explain and predict the observed properties of discrete systems that extend beyond simple power-law (Zipfian) relationships, (ii) it must be agnostic with respect to the types of pieces (tokens) of which discrete systems are composed, (iii) it must be agnostic with respect to mechanism, and (iv) it must be scale-independent. The mathematical theory that we describe and test in this study meets these criteria. It takes an approach that is both mechanism- and token-agnostic; we take the simplest measure of information, Hartley–Shannon, in which the tokens (also called signs or symbols in information theory) are explicitly without meaning [18–21] and embed this in a statistical mechanical framework which is ipso facto independent of mechanism. The theory distinguishes two types of discrete system: *heterogeneous*, in which the tokens are assembled sequentially in a *distinguishable* order; and *homogeneous* systems, in which tokens are assembled in an *indistinguishable* order. We show that the single differential equation that we derive, which embodies the principle of conservation of Hartley–Shannon information or CoHSI, accurately predicts the global properties of discrete systems (both heterogeneous and homogeneous) as diverse as proteins, computer software and digital music. The properties that are accurately predicted include the distinctly un-Zipfian size distributions that are seen identically in, for example, both proteins and software (figures 3 and 4) and that we will address in greater detail later in this article.

## 2. Heterogeneous discrete systems

Consider figure 1, a simple string of differently coloured beads appearing in order distinguishable by position. There are 35 beads altogether in 12 colours in this string, and an assemblage of 7 such strings of beads, as shown in figure 2 constitutes a simple example of a heterogeneous system. In our nomenclature, each bead is a token and each string of beads is a *component*. Simple though it is, numerous important discrete systems in both the natural and man-made world contain strings exactly like this. For example, this string could be a protein, where the beads are amino acids. Alternatively, it could just as easily be a particular computer program, where each bead represents a fundamental symbol of one of the hundreds of programming languages. It might also be a string of notes in a musical composition, this time with each bead corresponding to one note with different colours corresponding to different notes. Clearly, each representation will mean something only when we attach appropriate meaning to each bead. However, this does not alter the fundamental property that each of these disparate systems contains something very much like a string of beads as some kind of basic building block.

Table 1 illustrates with real-world examples the nomenclature we will use as we develop this argument. We consider a discrete system to be a set of *components*, each of which is built from a *unique alphabet* of discrete indivisible choices or *tokens* (also known as *symbols* or *signs* in information theory). At first glance, this seems a very coarse taxonomy. In the case of proteins, there is no mention of the domain of life or species or any other kind of aggregation. Similarly with computer programs, we do not include the programming language in which they were written or the application area that they serve. We will see that these considerations will turn out to be irrelevant. It might be thought that if systems as disparate as computer software, proteins and music share a

royalsociety publishing.org/journal/rsos    R. Soc. open sci. **6**: 191101

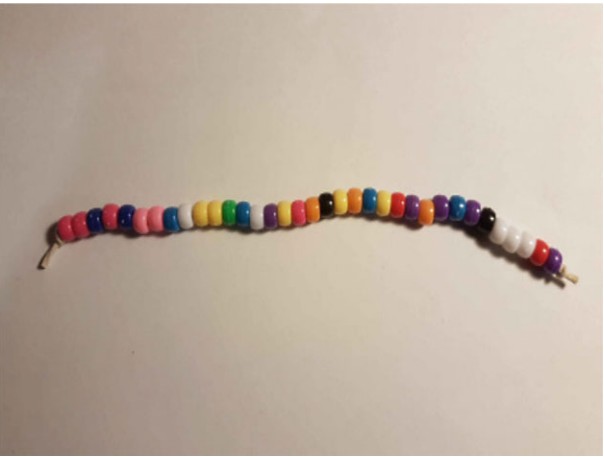

**Figure 1.** A simple string of coloured beads.

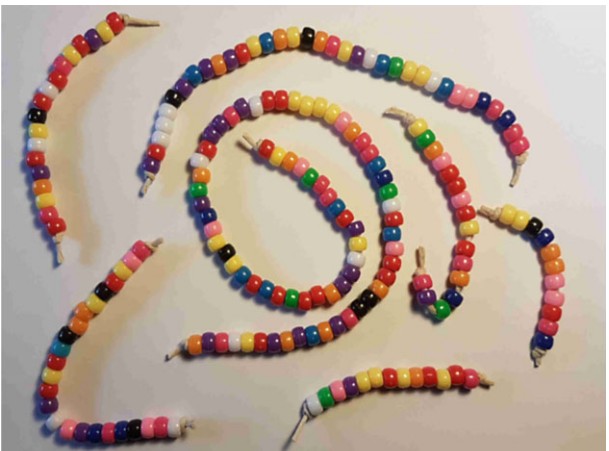

**Figure 2.** Illustrating the CoHSI heterogeneous model. Seven components are shown as strings of tokens that are distinguishable by both their colour and their order. In the case of the system of proteins, each string would correspond to a unique protein and each of the tokens would be an amino acid. Different colours would indicate different amino acids.

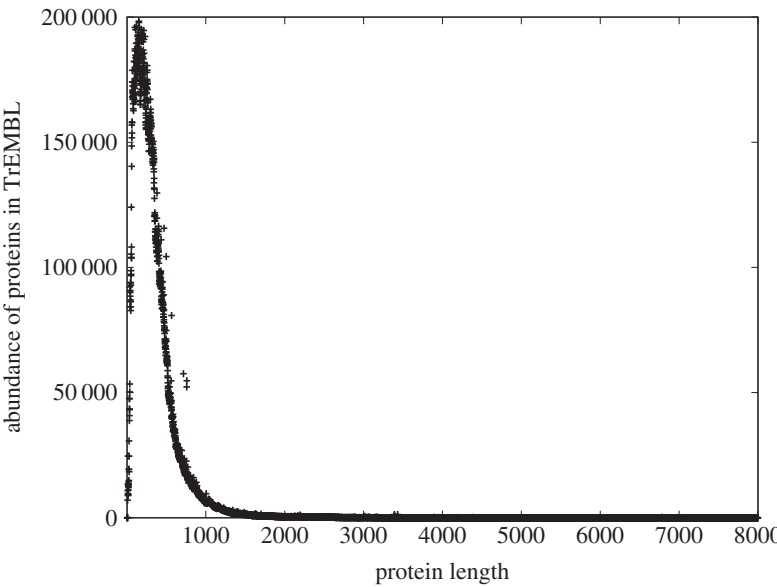

**Figure 3.** The frequency distributions of protein lengths measured in amino acids as represented in version 17-03 of the TrEMBL database, https:/uniprot.org/ totalling around 80.2 million proteins assembled from 26.9 billion amino acids.

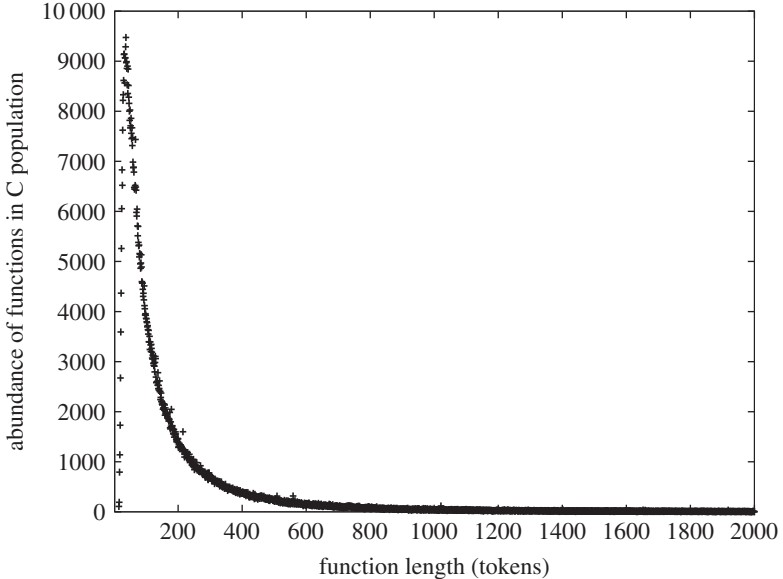

**Figure 4.** The frequency distributions of function lengths in 80 million lines of open-source software, in this case written in the programming language C, comprising some 500 million programming language tokens [12].

**Table 1.** Comparable entities in discrete systems considered in this study.

| system | component | token |
| --- | --- | --- |
| proteome | protein | amino acid |
| computer program | function | language token |
| music | composition | note |

fundamental organization equivalent to that of our simple string of beads, that these systems might also share other fundamental properties in common; this consideration is at the heart of this study.

When we define information content consistently across discrete systems, two measures of a component emerge naturally; these are the total length of the component and the size of the unique alphabet. If we consider the component (the string of beads) shown in figure 1, its length (in beads) is 35 and the size of the unique alphabet is 12 (there are 12 different colours of bead in this string). Similarly, if we consider a string of letters, say AABABFGAYXYTCM, then the length is 14 and the unique alphabet size is 9 (A,B,C,F,G,M,T,X,Y). Systems built in this way are ubiquitous. Some such as proteomes (a proteome is the full collection of proteins expressed by a species) and computer software packages are naturally so, whereas others such as music have moved from the analogue domain into the discrete domain with the advent of digital representations of music such as MusicXML [22]. Each type of system may be very different. For example, proteins and computer software are obviously and comprehensively different. While proteins are essential components of all known life forms, and it is generally accepted that they arose concomitantly with the origins of life some $3.5–3.8 \times 10^9$ years ago, computer programs are not (yet) essential to life on Earth and are the recent abstract product of deliberate cognitive activity on the part of humans. Music is a similar abstract product of deliberate cognitive activity, although its discrete digital representation is recent.

We now consider in more detail one of the component measures (length) that we defined above, and turn to figures 3 and 4, which show the distribution of component lengths for the two largest systems of table 1; the collection of all known proteins and a system of computer software. For example, version 17-03 of the protein database (https://uniprot.org/) contains more than 80 million entries, essentially the collection of all known proteins at the point of that release, each comprising strings of anywhere between 5 and some 37 000 amino acids, with each of the 22 genetically encoded amino acids represented digitally by a letter of the alphabet. If we plot the frequency distribution of protein lengths against their length, we obtain figure 3. Amongst other things, this shows that the proteins with length around 300 amino acids occur most frequently.

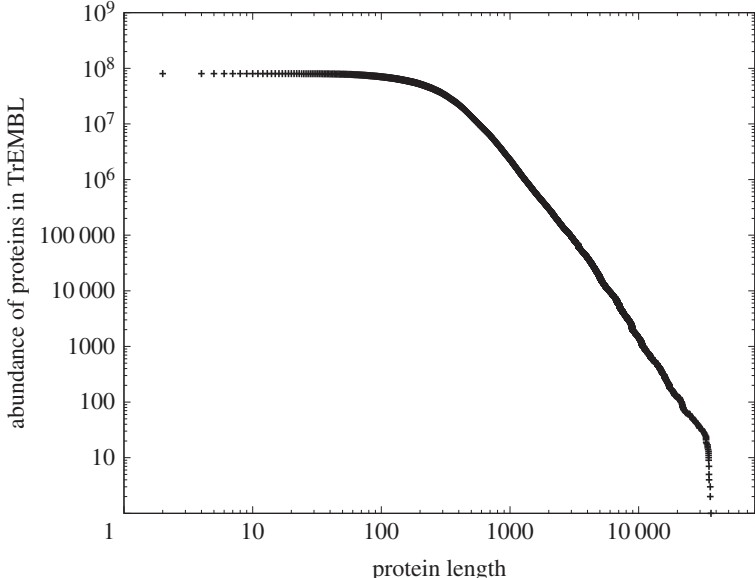

**Figure 5.** The data of figure 3, the frequency distributions of protein lengths, plotted as a complementary cumulative distribution function (ccdf).

Switching our attention to computer software, this consists of functions (components) built from strings of programming language tokens (syntactically indivisible symbols of a computer program such as **if**, **]** or **numberOfCollisions**). If we plot the frequency distribution of function lengths in a large quantity of open-source software from a typical Linux distribution against their length, we obtain figure 4.

From the point of view of this study, we need to know little more about the intrinsic nature of these systems, but they are clearly of completely different provenance; the proteins have arisen in the natural world in the course of the evolution of life, and computer software has arisen from human volition and the actions of human programmers. Despite these different origins of these systems, their distributions of component length show striking qualitative similarity—*is this a symptom of some deeper unifying principle?*

Pursuing our comparison of figures 3 and 4 a little further, we can see that although both systems are large the two systems are of quite different scale—by about a factor of 50 in tokens. Both have a sharply defined unimodal peak with almost linear slopes; both are significantly left-skewed and both appear to decay to zero for large component lengths in a similar manner. The nature of this decay can be understood better by plotting them as complementary cumulative distribution functions (ccdfs), utilizing the well-known noise-reducing properties of the ccdf [15].

It is immediately apparent that both figures 5 and 6 show the characteristic straight-line behaviour in log–log of a power-law tail over multiple decades, albeit with different slopes. Indeed in both cases, R reports an adjusted $R^2 > 0.99$ with a $p < 2.2 \times 10^{-16}$ with a slope of $-2.14 \pm 0.20$ in the case of figure 5 (over two decades) and a slope of $-1.52 \pm 0.08$ in the case of figure 6 (over four decades).

We might at this point start trying to fit different common statistical distributions to these data. Visually, they are similar to a gamma distribution, but gamma distributions have exponential tails, and neither figure 5 nor figure 6 shows any obvious evidence of this. In addition, the unimodal peak of figures 3 and 4 is unusually sharply distinguished. Alternatively, we might simply ascribe the similarity in their tails to the well-known ubiquity of power-laws [15] and look no further. However, we note that a value of an adjusted $R^2$ so close to 1, although a necessary condition to infer the presence of a power law, is not sufficient to infer such presence [23]. We will return to the issue of sufficiency but simply note here that a detailed analysis using the methods of Clauset *et al.* [23] outlined later, justifies us in describing the distributions of both figures 5 and 6 as having a power-law tail.

Although such justification is important, either of these strategies seems to limit our opportunities unnecessarily. The ubiquity of power laws is not well understood, but more importantly, *the majority of the contributing components in figures 3 and 4 fall into the unimodal peak (the power-law tail represents an interesting but minority adjunct, about 10% to 25% of the respective distributions).*

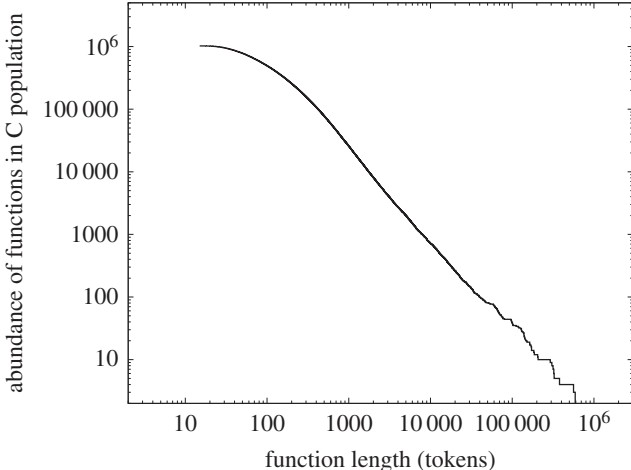

**Figure 6.** The data of figure 4, the frequency distributions of function lengths, plotted as a ccdf.

Instead, we will seek a more fundamental unifying principle in an attempt to understand why such disparate systems have such extraordinarily similar distributions. In doing so, we will try to understand the nature of the transition from the sharp unimodal peak to the power-law tail. *In short, we seek to derive the entire heterogeneous CoHSI distribution from first principles with as few assumptions as possible.*

We can draw clues from the intrinsic diversity of the two systems. If a unifying principle exists, it must have at least the following properties:

— It must necessarily be agnostic to the meaning of the symbols—programming language tokens and amino acids have absolutely no meaning in common.
— Since the two systems examined here evolved under such different external conditions, the unifying principle must be agnostic to any mechanism. We might of course make the assumption that diverse mechanisms will aggregate to the same result, but it would be more elegant to find a methodology in which mechanism is irrelevant and assumptions are dispensed with to the greatest extent possible.
— As we noted earlier, figures 3 ($\sim$2.7 $\times$ 10$^{10}$ tokens) and 4 ($\sim$6 $\times$ 10$^8$ tokens) represent significantly different scales. The principle must be scale-independent.

Similarities between physical systems are often associated with the unseen action of conservation principles. Conservation principles are well understood for such systems. For example, in Lagrangian systems, conservation principles are intimately associated with symmetries [24], and while scale is a form of symmetry, uncovering such principles is more challenging when we are considering not only physical but also biological systems and systems of origin in human cognition in which we do not have a Lagrangian.

Considering the first two points above, that an explanation must be independent of both mechanism and token meaning, the present approach significantly extends the initial development of theory (and supporting results) that dealt solely with the asymptotic power-law tail reported in [11,12,25] but omitted consideration of the sharp unimodal peak. It achieves this first by consolidating this asymptotic theory with work reported in arXiv preprints which showed how the solution could be naturally extended to include the unimodal peak and then completes the argument by placing it here on firm analytical grounds while embracing more sophisticated statistical tests for power-law behaviour due to Clauset *et al.* [23] for the extensive empirical support presented. The provenance is described in appendix A, *Consolidated arXiv work.*

Our approach embeds the original and arguably a simplest measure of information (Hartley–Shannon information) in a statistical mechanics framework. The rationale for this was that Hartley's original definition of information [18] *specifically cautions against assigning any meaning to the symbols.* Indeed, the only relevant consideration is that symbols are distinguishable. Furthermore, statistical mechanics does not consider mechanisms. Instead, it concerns itself with the overwhelmingly most likely distribution given certain constraints, irrespective of any mechanisms. It is not that such mechanisms are unimportant, they are simply irrelevant. From this, we were able to show with compelling support from measurement that this alone was sufficient to generate asymptotically, for long components in the tail of the distribution, the precise power laws observed not only in component length distributions but also

the distributions of the alphabets of unique tokens, for diverse systems [25]. However, as we noted above, to be sufficiently persuasive, *any candidate theory must be able to explain the more populous sharply defined unimodal peak as well as a power-law tail, preferably with no additional assumptions*.

The approach of combining information theory with statistical mechanics is not new—it has also notably been used by Frank [13], building on the maximum entropy framework of Jaynes [26], which is rooted in probability theory. Frank showed that by combining Shannon information [19] in a maximum entropy context along with additional information in the form of knowledge about, for example, the mean of a distribution, the common patterns of nature, such as Gaussian, power-law and exponential, naturally emerged as predicted by neutral generative processes.

We do not follow this general approach for two reasons.

First, although Frank is perfectly clear in that he is maximizing Jaynes' concept of *'information entropy'*; both *'entropy'* and *'information'* are often used synonymously in general literature, but they are not the same. Their conflation is famously attributed to John von Neumann by Claude Shannon [27]. It is true that they have the same mathematical form but they have entirely different units and entropy, von Neumann's comments notwithstanding, is perfectly well understood by engineers and physicists and furthermore is physically measurable [28]. We will have more to say on the distinction between (physical) entropy and information entropy shortly so we can avoid any use of the concept *'entropy'* in our main development as we feel it is more likely to obfuscate than to clarify the argument.

Second, some aspects of discrete systems are known not to conform to properties that can be represented by a single common pattern. We have already seen this with figures 3 and 4 which combine an unusually sharp unimodal peak abruptly transitioning into a power law.

To summarize, the questions addressed in this study are, first, whether or not the extraordinary similarity of figures 3 and 4 can be predicted mathematically without recourse to local mechanisms and without assigning meaning to the respective tokens or symbols used in the assembly of these systems. Second, does the theory derived by this approach also predict other observed scale-independent properties of a wide class of discrete systems, regardless of their natural origins or their development as the product of creative human endeavour? In the following sections, we build on prior work [11,12,25] embedding Hartley–Shannon information directly into statistical mechanics to show here that, without additional assumptions, and encapsulated in a single differential equation, the conservation of Hartley–Shannon information (or CoHSI) can indeed explain not only why figures 3 and 4 are so similar but also why other common patterns of behaviour, including Zipfian power laws, are characteristic of a wide variety of discrete systems.

This study therefore (i) develops a theory, (ii) makes predictions from that theory, (iii) collects substantial measurements from large datasets of different provenance, (iv) compares the properties of the experimental datasets with the predictions of the theory, (v) attempts but fails to falsify the predictions based on the experimental datasets, and (vi) in line with the recommendations of [29] and others provides the complete means including all software, computational framework, statistical analysis and data sources to reproduce the results independently.

# 3. Theory

## 3.1. Hartley–Shannon information and statistical mechanics

The methodology we use combines two disparate but long-established methodologies—statistical mechanics and information theory in a novel way using the simplest possible definition of information originally due to Hartley [18]. We will show that statistical mechanics can be used to predict the component distributions in general systems (such as those of table 1) made from discrete tokens subject to restrictions known as constraints.

The classical origins of statistical mechanics can be found in the kinetic theory of gases [30] (p. 217 *et seq*.) wherein constraints are applied by fixing the total number of particles and the total energy [31]. However, the methodology is very general and can equally well be used with different constraints on collections as disparate as those of table 1 where the only two constraints are total Hartley–Shannon information and the total number of tokens.

Hartley–Shannon information theory is the result of the pioneering works of Ralph Hartley [18] as developed later by Claude Shannon [19,20]. It forms the backbone of modern digital communication theory and is also astonishingly versatile.

*The Hartley–Shannon information content of a component in the sense we use here, is simply defined to be the natural logarithm of the total number of distinct ways of arranging the tokens of that component, without any regard for what those tokens actually mean* [21]. Token choice is equally likely and the *size* of the unique alphabet of each component is preserved as component contents are redistributed during the variational process.

## 3.2. Why conserve information?

In the kinetic theory of gases, conserving the total energy along with the total number of particles in the methodology of statistical mechanics leads directly to the Maxwell–Boltzmann distribution of particle velocities [31]. If we simply replace energy by information, it follows that another class of distributions will emerge related to information. These turn out to be length distributions. In retrospect, this is not surprising as the length of a sequence of symbols directly influences its information content. Indeed, following our earlier comments, the unique alphabet of symbols also influences the information content and we expect that it too will appear as a parameter in the resulting distributions.

The approach of conserving information is a perfectly reasonable thing to do as statistical mechanics is simply a mathematical technique—we require only that the conserved quantity be additive. If total energy is conserved, then it turns out that physical entropy necessarily intrudes. The origin of physical entropy is based on Clausius's work in the nineteenth century. In essence, conservation of energy is not enough to describe the phenomenon of irreversibility and resolve Loschmitz's paradox (macroscopic irreversible systems are built from microscopic reversible systems). Clausius and others resolved this by supplementing energy with an additive measurable quantity, entropy. By the Second Law of Thermodynamics, the physical entropy of a closed system can never decrease. When Boltzmann introduced his statistical mechanics approach, he *postulated* that entropy was proportional to the log of the number of possible configurations. Later Planck wrote this down as the famous $S = k \log W$, where $k$ is Boltzmann's constant and $W$ is the number of possible microstates of the system. This approach is therefore built on an assumption. **Either** we assume $S = k \log W$ **or** we can assume that the Lagrange parameter $\beta$, which determines the shape of the distribution, is equal to $1/kT_A$, where $T_A$ is the absolute temperature. *One implies the other* [31].

When we replace energy by information, we no longer need to be bound by these considerations. It so happens that Hartley–Shannon information content, like energy, is also additive for independent subsystems. The total energy $E$ of two independent subsystems with individual energies $E_1$ and $E_2$ is $E = E_1 + E_2$. Similarly by virtue of its logarithmic definition, the total Hartley–Shannon information content $I$ of two independent subsystems with individual information content $I_1$ and $I_2$ is $I = I_1 + I_2$.

It is therefore in the following sense that we assert that conservation of Hartley–Shannon information underlies the length distribution and other properties of discrete systems whatever their provenance. *It is a natural consequence of statistical mechanics that if we are presented with a system with a total number of tokens $T$ and a total information content $I$ distributed among its components, then the length distribution of those components is (as we will demonstrate below) overwhelmingly likely to be the size distribution we have already seen as figures 3 and 4. The scale-independence of the results follows from the fact that the shape of the predicted distribution is independent of $(T, I)$.*

## 3.3. Heterogeneous CoHSI equation

The classic formulation of statistical mechanics [31] for a simple heterogeneous system of the form of figure 2 when we set as constraints the total information content ($I$) and the total number of tokens ($T$), and using Stirling's approximation for $\log (t_i!)$ reduces to

$$\log \Omega = T \log T - T - \sum_{i=1}^{M} \{t_i \log (t_i) - t_i\} + \alpha \left\{ T - \sum_{i=1}^{M} t_i \right\} + \beta \left\{ I - \sum_{i=1}^{M} I_i \right\}, \tag{3.1}$$

where $\Omega$ is the number of ways of arranging the system (i.e. possible microstates), $\alpha$ and $\beta$ are Lagrange multipliers, $M$ is the number of components (represented as strings here), $t_i$ is the number of tokens in the $i$th string, $a_i$ is the unique alphabet (i.e. the number of unique tokens as defined by colour in figure 2) in the $i$th string, $I_i = I_i(t_i, a_i)$ is the Hartley–Shannon information content of the $i$th string, $T = \sum_{i=1}^{M} t_i$ is their total content of tokens and $I = \sum_{i=1}^{M} I_i$ is their total Hartley–Shannon information content. $\alpha$ controls the constraint on total size $T$ and $\beta$ controls the constraint on total information $I$. In essence, the variational

process envisages varying only the contents $t_i$ of each of the strings subject to $T$, $I$ and the $a_i$ remaining constant, until a maximum of $\log \Omega$ is found. Keeping the $a_i$ constant during this process is exactly analogous to keeping the energy levels constant in the development of the Maxwell–Boltzmann distribution [31]. The maximum is found by taking $\delta(\log \Omega) = 0$ (analogous to finding maxima in differential calculus). Following our earlier comments, we do not attempt to interpret this in terms of any kind of entropy; the process is simply finding the most likely distribution (which corresponds in the language of statistical mechanics to the macrostate with the greatest number of contributing microstates, where each microstate is assumed equi-probable).

Applying the $\delta()$ variation to (3.1) in the usual way and simplifying gives

$$0 = - \sum_{i=1}^{M} \left( \log t_i + \alpha + \beta \frac{dI_i}{dt_i} \right) \delta t_i. \tag{3.2}$$

The size of strings $t_i$ in a heterogeneous system can vary enormously, and at the largest sizes typically exceeds the maximum unique alphabet by many decades. For example, the size of proteins in the databases ranges from as few as 5 to over 37 000 amino acids, while the maximum unique alphabet of genetically encoded amino acids is 22. Thus, it is clear that to compute the Hartley–Shannon information content of a string, the distribution of tokens among strings needs to be considered separately for the situations where string length greatly exceeds its unique alphabet of tokens, and the situation where the size of the string is closer to the size of its unique alphabet.

Considering first what happens when strings are very large compared with their unique alphabet, i.e. $t_i \gg a_i$. In this case [12,25], the Hartley–Shannon information content is

$$I_i = \log (a_i \times a_i \times \cdots \times a_i) = \log (a_i^{t_i}) = t_i \log a_i. \tag{3.3}$$

In other words, for the example of a string of beads, in assembling the string, we select a bead $t_i$ times from a choice of $a_i$ colours of bead secure in the knowledge that since $t_i \gg a_i$, it is very unlikely that any of the $a_i$ colours would be missed out and **we therefore meet the requirement of having** *exactly $a_i$ unique* **colours** (i.e. at least one of each).

In these circumstances, (3.2) and (3.3) reduce to

$$0 = - \sum_{i=1}^{M} (\log t_i + \alpha + \beta \log a_i) \delta t_i. \tag{3.4}$$

Since this must be true whatever choices of $\delta t_i$ are made during their variation, the parenthesized term must be identically zero, which gives a pure power-law probability distribution function (pdf) with solution $t_i \sim a_i^{-\beta}$, or equivalently $a_i \sim t_i^{-1/\beta}$. This is why both length and unique alphabet distributions in heterogeneous systems asymptote to a power-law tail [25]. However, for the shorter strings of beads in figure 2, there is an increasingly high probability that we might miss out one of the colours available in the maximum unique alphabet as the $t_i$ beads are selected, breaking the fundamental assumption of the variational process that the $i$th string must retain a unique alphabet of **exactly** $a_i$.

We must therefore think more carefully how the Hartley–Shannon information content of shorter strings of beads is calculated in order to guarantee that each of the $a_i$ colours appears at least once among the $t_i$. This is not trivial, although its asymptotic nature when $t_i \gg a_i$ is already known from above.

To explore this when $t_i \sim a_i$, suppose we have a simple example of a string of $t_i = 5$ beads such that it contains *exactly* $a_i = 2$ different beads of colours A and B. The total number of ways this can be done if all combinations are equally probable $N(t_i, a_i)$, is given by

$$N(5, 2) = \frac{5!}{1!4!} + \frac{5!}{4!1!} + \frac{5!}{3!2!} + \frac{5!}{2!3!}. \tag{3.5}$$

Note:

— The first term on the right-hand side of (3.5) is the total number of ways of selecting 5 beads by using 1 bead of colour A and 4 beads of colour B. This is equal to 5 (ABBBB, BABBB, BBABB, BBBAB, BBBBA).
— The second term corresponds to 4 beads of colour A and 1 of B and is also equal to 5 (BAAAA, ABAAA, AABAA, AAABA, AAAAB).
— The third term corresponds to taking 3 beads of colour A and 2 of colour B. This is equal to 10 (AAABB, AABAB, AABBA, ABAAB, ABABA, ABBAA, BBAAA, BABAA, BAABA, BAAAB).

— The fourth term corresponds to taking 2 beads of colour A and 3 of colour B. This is also equal to 10 (BBBAA, BBABA, BBAAB, BABBA, BABAB, BAABB, AABBB, ABABB, ABBAB, ABBBA).

There are no other ways of arranging the string such that there are exactly two colours of bead and exactly five beads altogether. There are therefore $5 + 5 + 10 + 10 = 30$ different such strings in total. (Note that if we use the $t_i \gg a_i$ form for information content, we erroneously get a higher value $a_i^{t_i} = 2^5 = 32$, because this includes AAAAA and BBBBB, both of which violate $a_i = 2$.) *Looking ahead, it is this subtle reduction in information content in smaller strings which is responsible for the unimodal peak in figures 3 and 4.*

The denominators of (3.5) correspond to elements of the *additive compositions* (https://en.wikipedia.org/wiki/Partition_(number_theory) (accessed 4 June 2019)) of size 2 of the number 5. These are

$$5 = 1 + 4; \, 5 = 4 + 1; \, 5 = 3 + 2; \, 5 = 2 + 3. \tag{3.6}$$

There are other additive compositions such as $2 + 2 + 1$, but this corresponds to three different kinds of token, i.e. $a_i = 3$, so must be excluded.

It turns out that there is a recursive solution for $N(t_i, a_i)$. First, we slightly modify the definition in (3.5) by letting $N(t_i, a_i; a_i')$ be the number of ways of producing a string with $t_i$ beads containing exactly $a_i'$ unique colours chosen from a total unique number of colours of $a_i$ for that string, where $a_i' \leq a_i$. In this notation, for example, $N(5, 2; 1) = 2$ (AAAAA,BBBBB), $N(5, 2; 2) = 30$ (see above), $N(5, 3; 3) = 150$, $N(5, 4; 4) = 240$, $N(5, 5; 5) = 120$. The distinction between $a_i$ and $a_i'$ is to make way for the use of recursion and we note that by definition,

$$N(t_i, a_i) \equiv N(t_i, a_i; a_i).$$

It can be verified that the following recursion then generates the desired total number of ways $N(t_i, a_i; a_i)$ of generating a string of $t_i$ tokens from a unique set of tokens $a_i$.

> **for** $t_i = 1, \dots, t_i(MAX)$ **do**
> **for** $a_i = 1, \dots, t_i$ **do**
> $N(t_i, 1; 1) = 1;$
> **for** $i = 1, \dots, (a_i - 1)$ **do**
> $N(t_i, a_i; i) \leftarrow {}^{a_i}C_i N(t_i, i; i)$
> **end for**
> $N(t_i, a_i; a_i) \leftarrow a_i^{t_i} - \sum_{i=1}^{a_i - 1} N(t_i, a_i; i)$
> **end for**
> **end for**

We then arrive at the corresponding heterogeneous Hartley–Shannon information content for a string containing $t_i$ tokens chosen from a unique alphabet of $a_i$ tokens, *in which each of the $a_i$ appears at least once*, which is

$$I_i = \log\left(N(t_i, a_i)\right). \tag{3.7}$$

From (3.2), we therefore assert that the length distribution of a heterogeneous discrete system such as aggregations of software systems or proteins, at all scales with total number of tokens $T$ and total Hartley–Shannon information $I$ is therefore the solution $(t_i, a_i)$ of the **implicit** pdf (probability distribution function) for all $t_i$ such that Stirling's approximation is valid, corresponding to the solution of

$$\log t_i = -\alpha - \beta\left(\frac{\mathrm{d}I_i}{\mathrm{d}t_i}\right), \tag{3.8}$$

with

$$T = \sum_{i=1}^{M} t_i \tag{3.9}$$

and

$$I = \sum_{i=1}^{M} I_i = \sum_{i=1}^{M} \log N(t_i, a_i). \tag{3.10}$$

We note that the fact that (3.8) defines an implicit pdf is not without precedent [32,33]. Furthermore, the fact that it asymptotes to an explicit power-law pdf for $t_i \gg a_i$ as in (3.4) allows us to assert that (3.8) does indeed represent a pdf.

Before applying this theory predictively to various systems so that we may test it, we note that the Lagrange multipliers $\alpha$, $\beta$ in (3.8) are undetermined by the methodology of statistical mechanics but have the following interpretation:

— In the Boltzmann development constraining total energy, $\alpha$ parametrizes the total size of the system and therefore emerges naturally as a normalization condition so that a pdf in particle velocities results. Its role when constraining total information as in (3.8) is more subtle as the implicit nature means it can also affect the shape of the resulting length distribution.
— $\beta$ parametrizes the total payload and affects the shape of the distribution. The payload in our theory is Hartley–Shannon information, which depends on the size of the alphabet we use to categorize each of the components of a discrete system. Small alphabets correspond to large $\beta$ and vice versa. The implication of this indeterminism is that the range of values of $\beta$ which emerge via information theory can be rather different to those tied to physical systems. (In the Maxwell–Boltzmann distribution where the payload is energy, we recall that it is defined by Boltzmann's constant together with the absolute temperature, through the Second Law of Thermodynamics.)

## 3.4. The homogeneous CoHSI distribution

In contrast to the string of beads model of the heterogeneous case, there is another way of arranging our beads among components. We therefore define homogeneous systems here as systems wherein a component has only one kind of distinct token and the tokens are assembled in *indistinguishable* order, with each distinct token unique to one component, as shown for the example of coloured beads in figure 7. The indistinguishable order suggests that we can consider each component simply as a *bin*. This distribution encompasses a wide class of systems as different as word counts in textual documents and the distribution of elements in the universe. In such systems, a heterogeneous definition of information is degenerate and a different definition is necessary, which as we will show leads directly in our theory to an alternative proof of Zipf's Law which is known to be present in many datasets [15].

We anticipate that these distinctions between heterogeneous and homogeneous systems will lead to different information measures with consequently different properties. We should not be surprised by this as precisely the same occurs in physical systems where distinguishable order leads to Bose–Einstein statistics and indistinguishable order leads to Fermi–Dirac statistics [31].

Whichever definition of Hartley–Shannon information is used, the methodology simply tells us the most likely, or *canonical* distribution for discrete systems with the same fixed size and fixed information content, howsoever defined.

In homogeneous systems, each token carries a payload such that each bin contains *only tokens with the same payload, unique to that bin*. We represent this by assembling beads of the same colour in the appropriate bin (figure 7). We cannot simply set $a_i = 1$ as in the heterogeneous formulation as the total Hartley–Shannon information would then degenerate to zero. Recall, however, that we are only looking for the total number of ways of arranging the beads in these bins so that each bin has beads of a unique colour without any regard to order.

Suppose then we have $M$ bins such that the $i$th bin contains $t_i$ beads of unique colour $b_i$, where the total number of beads is $T = \sum_{i=1}^{M} t_i$. We will renumber them without loss of generality so that $t_1 \leq t_2 \leq \cdots \leq t_M$.

We proceed as follows. Select any bin and then fill it by selecting the $t_M$ beads of the corresponding colour. Since we are selecting from $M$ different colours, the probability that we will achieve this selecting at random is $(1/M)^{t_M}$. For the second bin, we then have an alphabet available of $M-1$ colours, so the probability of filling this bin with the $t_{M-1}$ beads of this colour from the remaining colours is $(1/(M-1))^{t_{M-1}}$, and so on.

The total number of ways $N_h$ this can be done is then given by this probability multiplied by the total number of ways in which $T$ beads can be selected without constraint, which is $T!$

$$N_h = T! \left[ \left( \frac{1}{M} \right)^{t_M} \times \left( \frac{1}{M-1} \right)^{t_{M-1}} \times \cdots \times \left( \frac{1}{1} \right)^{t_1} \right] = T! \prod_{i=1}^{M} \left( \frac{1}{i} \right)^{t_i}. \tag{3.11}$$

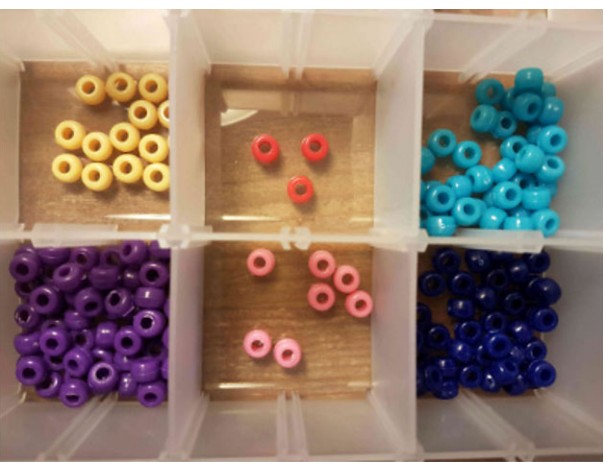

**Figure 7.** Illustrating the CoHSI homogeneous model. In each bin, all of the tokens are identical (i.e. of the same colour). Each bin contains tokens of a different colour and the beads are by definition in no distinguishable order.

Rewriting (3.11) then, the information content of this system is

$$\log N_h = \log T! + \sum_{i=1}^{M} t_i \log\left(\frac{1}{i}\right) = \log T! - \sum_{i=1}^{M} t_i \log i. \tag{3.12}$$

Following the heterogeneous development by folding this into the third term on the right-hand side of (3.1) and applying the $\delta()$ operator using Stirling's approximation, the equivalent of (3.2) now gives

$$0 = -\sum_{i=1}^{M}(\log t_i + \kappa + \eta \log i)\delta t_i, \tag{3.13}$$

leading to a homogeneous system pdf given by

$$t_i \sim i^{-\eta}, \tag{3.14}$$

where $\eta$, $\kappa$ are once again Lagrange undetermined multipliers.

There are notable differences between this and the heterogeneous case:

— No approximation is necessary for components (i.e. bins rather than strings as there is no distinguishable order), for which $t_i$ is comparable to $a_i$, since this does not arise in the homogeneous case.
— This is a pure power law at all values of $t_i$ but arranged in order of *rank*; it is in fact Zipf's Law, except for bins which have the lowest populations, in which case Stirling's Law may not be sufficiently accurate and the result is a natural droop in the tail of the distribution, i.e. the most sparsely populated bins, as we shall see.

# 4. Results

## 4.1. Solution of the heterogeneous CoHSI equation

We recall that (3.8) involved the use of Stirling's approximation to $\log(t_i!)$ but we could use higher-order approximations such as that of Ramanujan [34]. Unlike the homogeneous case which follows, this was not, however, necessary as the essential features of the unimodal peak emerge before Stirling's approximation becomes unacceptable for $t_i$ close to the origin; thus improving the approximation does not add anything fundamental to the heterogeneous case. We can solve (3.8) either in its differential form or in an integrated form. Here we use the differential form.

There are three practical difficulties:

— The function $N(t_i, a_i)$ can indeed be evaluated recursively using integer values of $t_i$, $a_i$. However, to solve an implicit equation, we need to analytically extend the domain over which this function is defined, to the real numbers. Since it involves sums of terms with pure factorials, which could in

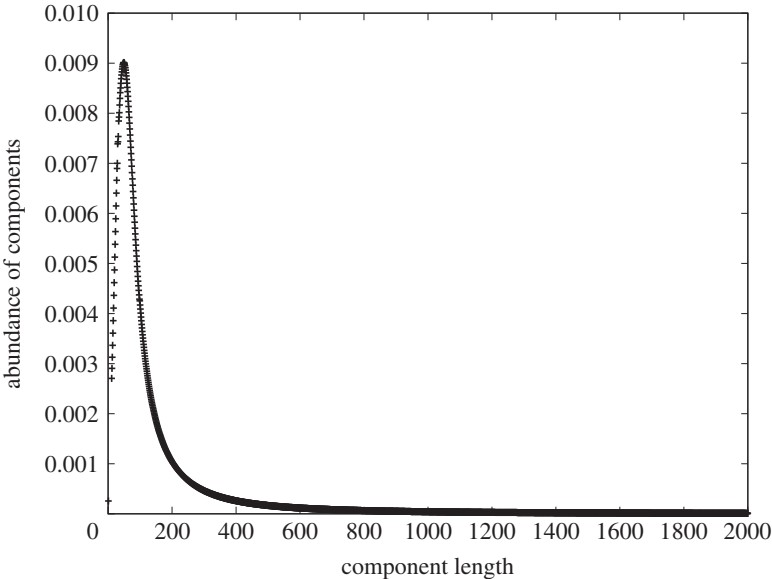

**Figure 8.** Illustrating a typical CoHSI heterogeneous solution. This naturally produces the sharp unimodal peak transitioning into the power-law tail.

principle be written out in full explicitly by unwinding the recursion, it is sufficient to replace the factorials with the gamma function identity, $n! = \Gamma(n + 1)$. The gamma function does the necessary continuation in a natural way. The real-valued $N(t_i, a_i)$ is then coincident with the integer-valued $N(t_i, a_i)$ at integral values of $t_i, a_i$.

— We must then differentiate $\log(N(t_i, a_i))$. The above continuation allows us to do this in practice by evaluating $\log(N(t_i, a_i))$ over an integer grid in $t_i, a_i$, and then calculating the differential coefficient by finite differences and interpolation on this grid.

— The function $N(t_i, a_i)$ can only be evaluated accurately for a relatively small range of integers with conventional arithmetic (approximately, $t_i, a_i \in [1, 30]$, depending on machine arithmetic precision). Fortunately, the solution asymptotes into the power-law tail quite quickly and even this rather restricted range proved more than enough to enclose the unimodal peak, as using unrestricted precision arithmetic slows computation down unacceptably.

Once we did this, a typical solution of (3.8) looks like figure 8 ($\alpha = 6$, $\beta = 0.5$).

This is qualitatively identical to figures 3 and 4. It is illuminating to see how and when the solution departs from the power-law tail as we approach the origin from large $t_i$. Figure 9 gives a compelling demonstration of this. Analysing this departure, it is caused by the information function $\log(N(t_i, a_i))$ reducing and flattening more quickly than the asymptotic information function $t_i \log(a_i)$ because of the constraint to maintain the unique alphabet $a_i$ at the same value.

This solution is explored for different values of the Lagrange parameters $\alpha$, $\beta$ in [35]. In essence, both Lagrange parameters contribute to the shape and position of the unimodal peak because of the implicit nature of the solutions and there indeed may be no solutions for the smallest values of $t_i$ for $\alpha$, $\beta$. This is certainly the case with the datasets we used—for example a total of only six sequences shorter than seven amino acids is included in the TrEMBL 19-04 database and the smallest legal component in the programming language C, for example, contains three tokens, but it exists only as an oddity as it has no functional behaviour. Whilst fitting the asymptotic power law is relatively simple, the implicit nature of (3.8) and other complications, such as the scale of the numerics and the subtle relationship between $\alpha$, $\beta$, the alphabet of tokens and corresponding parameters in the real data as revealed in [35], present significant challenges for the quantitative fitting of our real datasets to the shape and position of the mode in the CoHSI heterogeneous equation and this remains a work in progress.

We consider this to be an ample qualitative demonstration of the properties of the CoHSI heterogeneous equation (3.8) and is a validation of our mechanism- and token-agnostic approach. We will demonstrate this in depth shortly with numerous examples of real data, but first, we will complete this section by outlining how the no less interesting homogeneous equation may be solved.

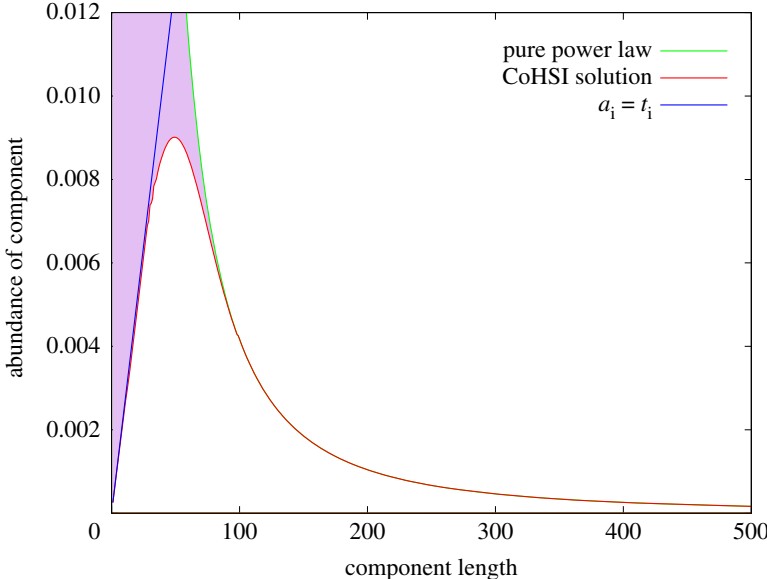

**Figure 9.** A close-up of figure 8 illustrating how the pure power law (green) is coincident with the full CoHSI solution (red) for component lengths above around 80 tokens but quickly departs from the full CoHSI solution for smaller components before abruptly reaching its peak value. The blue line indicates the natural boundary condition $t_i = a_i$, since manifestly no component can be smaller than its unique alphabet, which is of course enforced by the recursion. The purple region is the area between the pure power law and the full CoHSI solution.

## 4.2. Solution of the homogeneous CoHSI equation

Solving the homogeneous form was much less challenging because it is explicit; however, this equation too has an interesting property. We stated earlier that the homogeneous CoHSI equation serves as a proof of Zipf's Law, an empirical law of great ubiquity. However, Zipf's Law is a pure power law, leading to a straight line when plotted as frequency vs rank in log–log on a ccdf. Indeed this straight line, usually over several decades, is used as the fingerprint of Zipf's Law when analysing real data.

However, for the highest ranks (i.e. the bins with sparsest occupancy), a droop is often observed indicating these bins are less well populated than would be expected with a pure power law. Zipf's law does not allow for this drooping tail but it would be supportive of our approach if such a droop naturally emerged from the mathematics. In short, it does and the source of the droop is the gradually increasing departures between Stirling's approximation and the approximated function $\log(t_i!)$ leading up to (3.13), when bins are sparsely populated, i.e. $t_i$ is small. Recall in the heterogeneous case, the equivalent approximation in the solution occurs for the lowest values of $t_i$ before the heterogeneous unimodal peak has appeared and including better approximations did not add any new features of interest in the frequency vs length distribution. In the homogeneous case, however, the data are ordered as frequency vs rank and the sparsely populated bins appear in the tail of this distribution. In this case, including the higher-order approximations (e.g. [34] or even an exact evaluation for the smallest $t_i$) leads naturally to a new feature. The tail of the otherwise pure power law corresponding to Zipf's Law, droops. This can clearly be seen in figure 10, where we compare the Stirling and Ramanujan approximations and the exact evaluation in the tail of the distribution.

We would argue that the natural appearance of this droop from the mathematics alone is further supporting qualitative evidence for our approach, although we must inject a note of caution in interpreting this in real data, as the affected bins will be sparsely populated by definition and subject to considerable statistical fluctuation. The best we might expect is a tendency for tails to droop in real-world datasets.

## 4.3. Testing the predictions of CoHSI

So far, we have presented the theory behind the heterogeneous and homogeneous CoHSI models and how they may be solved. We have also seen that the CoHSI heterogeneous prediction of length distributions is in close agreement with those observed in systems as disparate as collections of

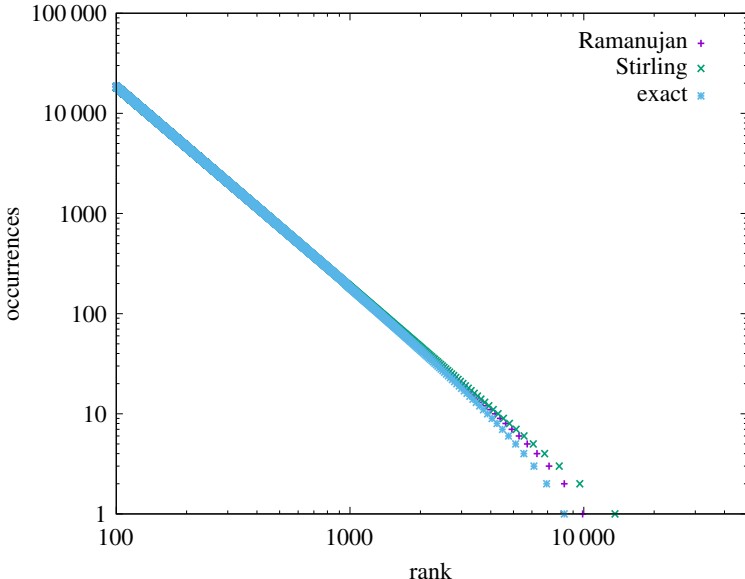

**Figure 10.** Illustrating a typical CoHSI homogeneous solution for three different approximations to the log factorial. As can be seen, handling the sparsely populated bins correctly leads to a natural droop in the predicted power law.

proteins and computer software, providing a bridging but parsimonious theory to explain why they are so extraordinarily alike. Our approach is therefore theory-driven with prediction rather than data-driven with assumptions about qualifying standard distributions.

Moreover, the existence of a CoHSI theory embracing *both* heterogeneous and homogeneous discrete systems considerably increases the number of possible cases in which the predictions of CoHSI can readily be tested.

This we now do.

### 4.3.1. Justifying the presence of power-law tails

Since both heterogeneous and homogeneous CoHSI solutions asymptote to a power law, it is appropriate to return to a consideration of whether or not the datasets analysed in figures 5 and 6 indeed display power-law tails.

As mentioned earlier, an adjusted $R^2$ fit close to 1 in a linearity test on a log–log scale [23] is certainly necessary but is not of itself sufficient to conclude power-law behaviour. There are a number of facets to this subtle statistical issue. Can we reject a power law on well-defined statistical grounds and if not, would a different distribution such as lognormal provide a better fit? Our strategy (which follows below) to address this issue is defined by our methodology; we are working from first principles with a consistent theory, the predictions of which we attempt to falsify by examination of experimental datasets. Whenever something as apparently ubiquitous as a power-law distribution appears, it is tempting to enter the long-standing debate about whether or not it is in fact a power law or whether lognormal, stretched exponential or some other distribution might give a better fit. However, as Clauset *et al.* [23] advise, '*In cases such as these, it is important to look at physical motivating or theoretical factors to make a sensible judgement about which distributional form is more reasonable—we must consider whether there is a mechanistic or other non-statistical argument favouring one distribution or another.*'

We concur completely. In this case, we have:

— A model with both physical and theoretical motivating factors.
— The power law is only the asymptotic and indeed minority part of the CoHSI heterogeneous distribution. The CoHSI distribution also explains the sharply defined unimodal peak without any further assumption.
— The adjusted $R^2$ are extremely close to 1 in each of the highly disparate datasets (computer software and proteins) that we examined.

It is therefore sufficient for our arguments to demonstrate rigorously that power-law behaviour in the tails of figures 5 and 6 is *not* ruled out. This process relies on sophisticated likelihood arguments

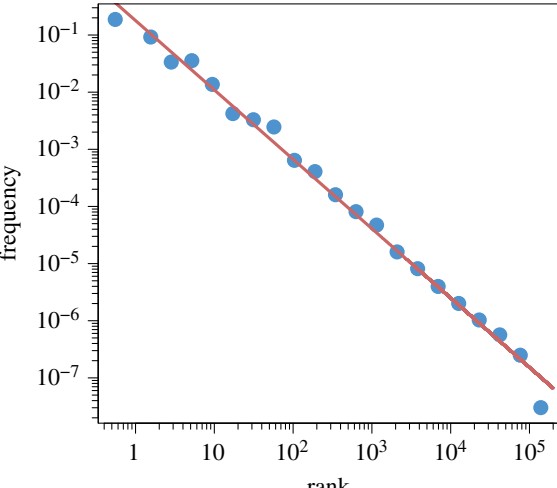

**Figure 11.** The data of figure 5 exponentially binned and with proteins with less than 500 amino acids (the unimodal peak) removed. A power law has been optimally fitted.

described in detail by Clauset *et al.* [23] and implemented in R as the *poweRlaw* package [36] in which Monte Carlo simulations are used to assess whether a particular distribution is possible. With our datasets for both proteins and software in their raw form, such simulations were inordinately expensive in computer time, so each of the datasets was first examined by exponential binning of the raw data to balance the bin contents followed by fitting to a power law. For the data of figure 5, the results of this process are exemplified by figure 11.

Visually, this is satisfactory, so the binning process was adapted to produce from the data of figures 5 and 6 suitable input to the Monte Carlo bootstrap process in the *poweRlaw* package. When the dataset of figure 5 was tested for the ability to reject a power law, the associated *p*-value was 0.789. The corresponding *p*-value for the dataset of figure 6 was 0.922. Clauset *et al.* [23] suggest a cut-off *p*-value of 0.1, *below* which a power law is not considered plausible. Therefore, we conclude that the presence of a power law in both of these datasets cannot be rejected.

Taking these results together, the fundamental nature of the CoHSI distribution and its central role in defining the length distributions of these highly disparate discrete datasets at all scales is strongly supported qualitatively by these results.

### 4.3.2. Simultaneous heterogeneous and homogeneous behaviour

CoHSI theory is consistent with the simultaneous presence of both heterogeneous and homogeneous behaviour in the *same system*; the two behaviours would then each correspond to a different but consistent categorization of the same data.

We test this prediction using the data for the systems of proteins and of computer software. The heterogeneous CoHSI theory predicted with considerable precision the unimodal peak transitioning into a precise power-law tail in both these systems (figures 3 and 5 for proteins and 4 and 6 for software). However, suppose that for both proteins and software we recategorize the very same data into a homogeneous system by simply redefining the '*frequency of occurrence of a specific length*' for proteins or software functions *as a bin*; we can then rank order the bins. This clearly satisfies the requirements of the homogeneous model and the contents of the bins are equally clearly in indistinguishable order—we simply do not care which proteins are in a particular bin, provided they have the appropriate length for that bin. In the process of rank ordering, we are discarding knowledge about the lengths.

Homogeneous CoHSI theory then states that when heterogeneous frequencies are ordered by rank, we should observe a Zipfian power law with a drooping tail. It might be argued that this is a trivial observation as we already know that both heterogeneous distributions have power-law tails. However, it should also be recalled that the majority of the components of these two systems (approx. 90% and 75%) appear in the unimodal peaks of figures 3 and 4, respectively, rather than in the power-law tail, so it is far from intuitively obvious how these should appear when arranged in rank order.

However, figures 12 and 13 illustrate very clearly that the prediction of CoHSI homogeneous behaviour for the rank-ordered heterogeneous distribution is accurate. The predicted straight-line power-law behaviour appears to be emphatically present over multiple decades in both systems, with

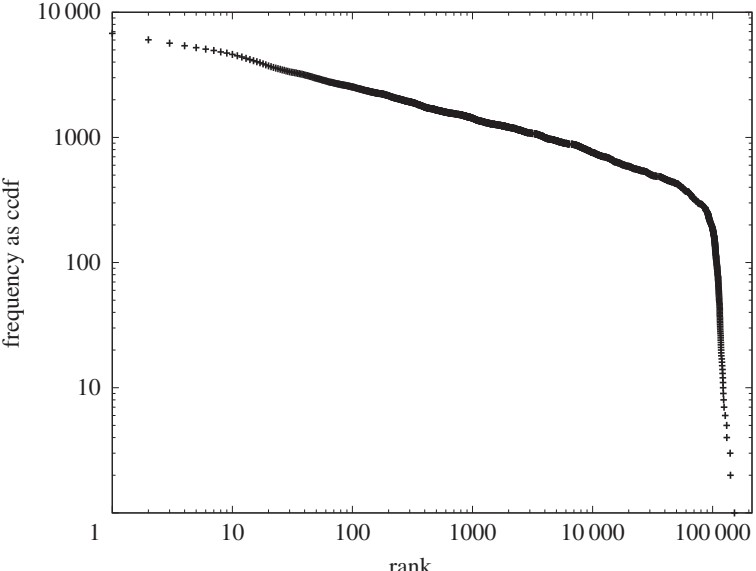

**Figure 12.** The heterogeneous distribution of protein lengths in figure 5 grouped as a homogeneous distribution and plotted as frequency *versus* rank.

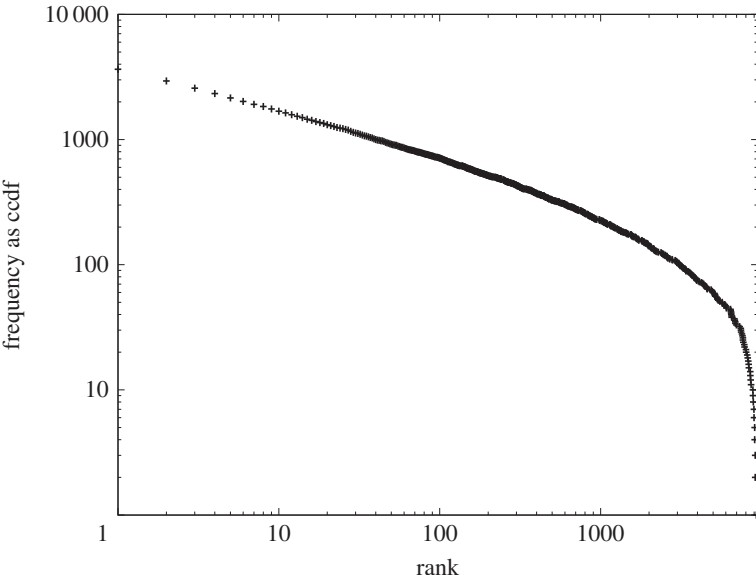

**Figure 13.** The heterogeneous distribution of software function lengths of figure 6 grouped as a homogeneous distribution and plotted as frequency *versus* rank.

the expected droop in the tail for the sparsely occupied bins. In both cases, R reports an adjusted $R^2 > 0.99$ with a $p < 2.2 \times 10^{-16}$ with a slope of $-0.27 \pm 0.04$ in the case of figure 12 (four decades) and a slope of $-0.43 \pm 0.05$ in the case of figure 6 (three decades). This necessary condition for a power law is therefore satisfied and further investigation using the bootstrap techniques of Clauset *et al.* [23], described earlier, gave *p*-values of 0.923 and 0.998 for these two datasets respectively. Once again, we conclude that we cannot reject a power law as the underlying distribution.

This predicted but non-intuitive behaviour provides further substantial experimental support for the CoHSI theory.

### 4.3.3. Categorization and the uniqueness of alphabets

When considering the application of the heterogeneous CoHSI model, a unique alphabet has to be defined consistently so that we can populate the components appropriately. It is not always obvious

how this can be done. Consider, for example, a colour-blind person counting the number of beads in strings of coloured beads and categorizing the beads by colour. They will count the same total number of beads as a person with normal colour-sight but they will probably categorize them differently by colour. Provided the normally sighted and colour-blind person are self-consistent, who has the correct categorization? Clearly, the total number of beads cannot depend on how the alphabet was categorized, and thus we must assume that both categorizations, although different, are equally valid. How would the co-existence of two different but consistent categorizations by the alphabet affect the predictions of CoHSI?

We will take as a concrete example digitally represented music, in which compositions are represented in terms of notes. Musical notes not only have a pitch but they also have duration. This raises the question as to whether we should distinguish, for example, between a short duration middle C and a long duration middle C. Provided we are consistent, we know it cannot affect the number of notes in the composition. However, CoHSI then implies a special relationship between two consistent but alternative alphabets categorizing the same system as we will now see.

We can use the asymptotic power-law behaviour of the CoHSI heterogeneous equation to derive such a relationship which we can then test. If we consider the 88 notes of a full-scale piano as defining the possible notes in the equal-tempered scale used in the vast majority of published music, then we have a candidate unique alphabet of up to 88, $a_i'$ say, the *(no-duration alphabet)*. However, we can subdivide this alphabet quite naturally and consistently into notes *and* duration. The standard durations are divided into fractions of a whole note as breve (2), semi-breve (1), minim (1/2), crotchet (1/4), quaver (1/8), semiquaver (1/16) and demisemiquaver (1/32). There are others defined off either end of this list but they are sufficiently rare that none occurred in our datasets. This gives seven durations of each note and expands the potential unique alphabet considerably up to $88 \times 7 = 616$ items, $a_i''$ say, the *(duration alphabet)*.

Since it cannot affect the size of each composition, then our asymptotic theory (3.4) tells us that

$$t_i \sim (a_i')^{-\beta'} \tag{4.1}$$

and

$$t_i \sim (a_i'')^{-\beta''}, \tag{4.2}$$

where $a_i'$, $a_i''$ are the two unique alphabets and $\beta'$, $\beta''$ their slopes. *We can see straight away from (4.1) and (4.2), that CoHSI predicts that the two unique alphabets must themselves be related asymptotically as a power law.*

$$(a_i')^{-\beta'} \sim (a_i'')^{-\beta''} \Rightarrow a_i' \sim (a_i'')^{-\beta'''}, \tag{4.3}$$

where $\beta''' = -\beta''/\beta'$. This implies that if we plot the size of the unique alphabet of a composition *including duration*, against the size of the unique alphabet of the composition *not including duration*, on a log–log scale, we should see a straight line. This is certainly not intuitive but is easy to test and the result for the whole population of digital music described in appendix A is shown as figure 14. R reports that the associated *p*-value matching the power-law tail linearity in the ccdf of figure 14 is less than $2.2 \times 10^{-16}$ over the duration range 10.0–616.0, with an adjusted *R*-squared value of 0.977. The slope is $1.58 \pm 0.09$.

This is indeed closely linear exactly as predicted by CoHSI even though the system (883 compositions) is statistically a much smaller sample than the populations of proteins and software considered elsewhere in this study. The bootstrap procedure of [23] gave a *p*-value of 0.351 so again we cannot reject the presence of a power law.

### 4.3.4. Scale-independence and the longest component

CoHSI heterogeneous and homogeneous theory is not dependent on scale. Both distributions are independent of the total size of the system in tokens $T$ and the total information content $I$. This scale-independence directly implies a profound property. *The largest component in a system as it grows is determined only by the total size of that system.*

In the case of proteins, these are strings of amino acids which fold in extraordinarily complex ways as part of their defining functionality. The mode of the length distribution of figure 5 is around 300 amino acids, and some 130 000 proteins have this length in this plot. Since each position could be occupied by up to 22 amino acids, the total number of possible ways of arranging 22 amino acids in 300 positions is $22^{300}$, a gigantic number. In other words, only an infinitesimal fraction of possible rearrangements $(130\,000/(22)^{300})$ has been explored by the known proteins. An important question to ask therefore is

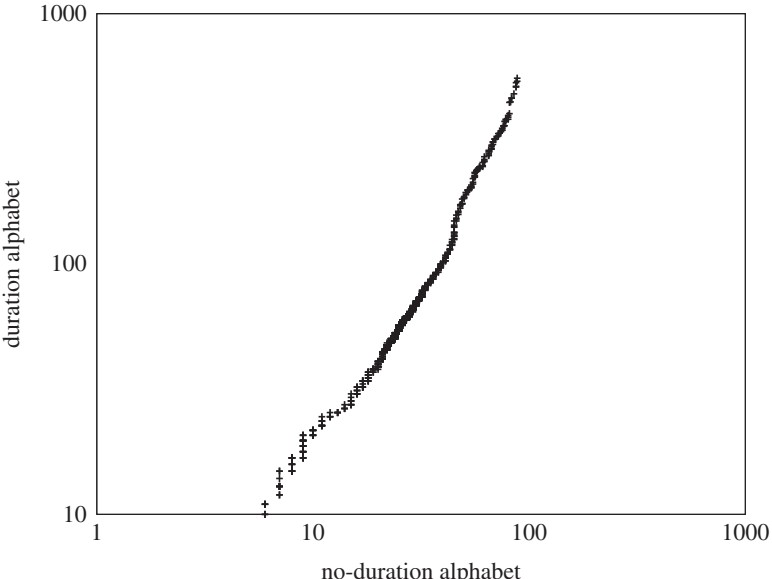

**Figure 14.** A comparison of the two alphabets, with and without note duration as log—log showing their clear linear power-law relationship. Each of the 883 data points corresponds to the size of the duration alphabet plotted against that of the no-duration alphabet for a single composition.

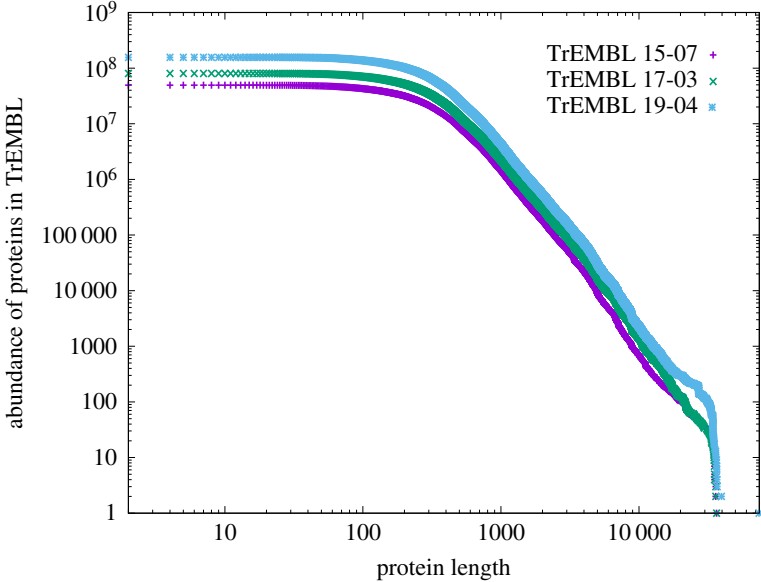

**Figure 15.** The cumulative ccdfs of three releases of the TrEMBL protein database spread across some four years, 15-07, 17-03 and 19-04. In this period, the database grew by a factor of four. The self-similarity is obvious just as it is in software systems [12].

why do proteins with over 36 000 amino acids occur in this collection? Surely nature does not need the implied degrees of freedom? CoHSI heterogeneous theory offers a simple solution to this question. *Very long proteins are inevitable; it is unnecessary to postulate that they result from natural selection.* We show the impact of scale-independence on the longest protein in a collection as figure 15. This shows three versions of the TrEMBL database over approximately 4 years (it is updated monthly), during which the database grew by a factor of four as can be seen on the *y*-axis. The strong self-similarity is obvious.

For example, by drawing a line upwards from protein length 10 000 on figure 15, it is clear that there are, by around a factor of 10, more proteins of this length in release 19-04 than in release 15-07. The tail of these distributions is statistically noisy, but while the largest validated protein in release 15-07 is 36 805 amino acids (Q3ASY8_CHLCH, *Chlorobium chlorochromatii* (strain CaD3)), in release 19-04, it is 39 677 amino acids (A0A410P257_9BACT, *Vampirococcus* sp. LiM). These results are consistent not just with the *number* of identified very long proteins increasing, but also their *maximum*

*length* increasing as the size of the collection grows. In fact, TrEMBL release 19-04 contains an uncharacterized protein of >74 400 amino acids (A0A316Q3J5_9FIRM *Clostridiales* bacterium) but its identification is still preliminary.

In the case of software, the occurrence of unusually large components (for the sake of argument say at least a factor of 20 bigger than the modal size) has hitherto been considered a failure of design and is generally associated with poor practice and increased potential for error [37–39]. However, CoHSI heterogeneous theory also applies to software as we have already seen earlier in figures 4 and 6, again with the clear implication that unusually long components are an *inevitable* by-product of the total size of the system. The importance of this conclusion is that software designers must then switch their software design techniques from avoidance of such long components, since they are essentially unavoidable, to mitigating any potential damaging effects they might have.

# 5. Conclusion

We make the following claims:

— Discrete systems as disparate, as the known collection of proteins, on the one hand, and computer software, on the other, share common system properties to an extraordinary degree. This is a matter of observation, and we single out, in particular, the distribution of lengths of their components. This similarity is evident in spite of their completely different provenance.
— A mechanism- and token-agnostic scale-independent theory embracing statistical mechanics in which the simplest possible measure of Hartley–Shannon information is embedded as a constraint (CoHSI) is capable of explaining this underlying similarity with only the standard assumptions of statistical mechanics; that all microstates are equally probable; and that components are reasonably well populated so that Stirling's approximation is satisfactory, (and even this latter was mitigated by higher-order approximations).
— Two forms of CoHSI system naturally emerge from the theory depending on how humans categorize the discrete systems they measure:
  (a) Heterogeneous systems in which each component is a string of tokens chosen in distinguishable order from some unique alphabet of tokens. These are characterized by a length distribution as shown in figure 8.
  (b) Homogeneous systems in which each component is a bin containing tokens in indistinguishable order and of a single kind that represents some shared property unique to that bin. These systems are characterized by a frequency *versus* rank distribution as shown in figure 10.
— We have demonstrated the accuracy of CoHSI predictions and some of its implications in a variety of both heterogeneous and homogeneous systems, including the known set of all proteins, large amounts of computer software and a body of digital music.
— By using the example of digital music choosing notes with and without information about duration, we have demonstrated that CoHSI heterogeneous theory predictions are unaffected by consistent but different choices of categories, i.e. alphabets. Moreover, the predicted but non-intuitive power-law relationship between different alphabets was confirmed as present.
— Although we incidentally give a proof of Zipf's Law in the case of homogeneous systems, the asymptotic power law present in heterogeneous systems is a minority feature. We stress that CoHSI theory predicts the entire distribution and notably the sharp unimodal peak abundantly obvious in figures 3 and 4, even though these systems differ in size by a factor of 50. (We should recall, however, that in spite of its minority status, the asymptotic power-law behaviour in the tail of a heterogeneous system implies the existence of unusually long components whose length depends only on the total size of the system. For example, we showed in multi-gigabyte collection of proteins, a number of proteins exist which exceed the average protein length by a factor of 100, and demonstrated that the longest known protein does indeed increase predictably with size of collection.)
— Last but not least, CoHSI theory predicts a previously unknown property of discrete systems such as proteins and software, whereby both heterogeneous and homogeneous behaviour concurrently manifests itself, simply depending on how information content is defined, as shown in figures 5 and 12. In other words, what we measure is determined by how we choose to categorize a system, providing we are consistent. We have yet to explore the full implications of this.

Finally, we reiterate that here we are neither explicitly data-fitting to known distributions, nor comparing known distributions. Instead, we have started from first principles guided by important

clues provided by the datasets themselves. These clues necessarily led us to a scale-independent, mechanism-independent and token-independent theory capable of making predictions. It so turns out that power laws are a natural asymptotic feature of our theory (given their ubiquity, we might worry if they were not), but this played a minor part in the theoretical development; it is an emergent property.

We have attempted to falsify the resulting predictions of our theory in large numbers of tests on systems big and small and of completely different provenance. We have so far failed to falsify these predictions, but it is important to note that CoHSI is not a straitjacket and embodies the ergodic nature of statistical mechanics. It is perfectly possible within our theory for a discrete system to have a different length distribution other than that predicted by CoHSI; CoHSI only identifies the overwhelmingly most likely distribution within its constraints. This can be seen clearly by construction in the domain of computer software. In a matter of seconds, the simple computer program shown in appendix A could be copied several million times and the result deemed a system in our nomenclature. Artificial though it may seem, this is a perfectly valid system in a theory which is agnostic with regard to the tokens—the functionality, however useless, is irrelevant. This system manifestly does not satisfy CoHSI but is simply unlikely in the ensemble of all systems with the same number of tokens $T$ and the same information content $I$. In the natural world too, natural selection could force a departure from the most likely distribution implied by CoHSI, which we interpret as CoHSI acting only to guide and not to enforce. We therefore conclude the following:

> If presented with a discrete system containing $T$ tokens in all, and an overall Hartley–Shannon information content $I$ (defined appropriately but consistently according to whether it is heterogeneous or homogeneous), it is overwhelmingly likely to have the canonical heterogeneous or homogeneous length distribution predicted by CoHSI.

We have presented a substantial set of data examples to justify this.

Data accessibility. This study adheres to the transparency and reproducibility principles espoused by [29,40–44] and includes references to all methods and source code necessary to reproduce the results presented. These are referred to here as the *reproducibility deliverables* and are available at https://datadryad.org/stash/dataset/doi:10.5061/dryad.gm957n1 and also at https://www.leshatton.org/HattonWarr_RSOS_Jun-2019.html. Each reproducibility deliverable allows all results, tables and diagrams to be reproduced individually for that study, as well as performing verification checks on machine environment, availability of essential open-source packages, quality of arithmetic and regression testing of the outputs [45].
Authors' contributions. L.H. performed the analyses, L.H. and G.W. developed the arguments, discussed the results and contributed to the text of the manuscript. Both authors gave final approval for publication.
Competing interests. We declare we have no competing interests.
Funding. We received no funding for this study.
Acknowledgements. We are very grateful to Bob Chapman and Andy Mount who provided invaluable feedback on this manuscript. We are also grateful to two anonymous reviewers who provided comments that substantially clarified the manuscript.

# Appendix A

## A.1. Consolidated *arXiv* work

This study consolidates early peer-reviewed work on the asymptotic power-law tail present in the heterogeneous distribution [11,12,25] with later results from a series of *arXiv* papers, and then extends this further as follows. The methods of the solution proposed for the differential form of the full theory presented in [46], which includes treatment of the unimodal peak, are here placed on firm analytical grounds whereby the solutions may be extended from integer values to the real line. (Figures 1–7 and a version of table 1 also appeared in this preprint.) In addition the results have been exposed to more sophisticated statistical tests due to [23], to allow us to assert that asymptotic power-law behaviour is indeed plausible in each of the datasets we use here. Figures 8 and 9 in the present publication, with their underlying analysis, appeared in [35] and figure 10 in the present publication is in [47]. Finally, it is appropriate to mention here that as this is a consolidated work over several years, numerous colleagues contributed at early stages and these are individually acknowledged in each *arXiv* paper as appropriate.

## A.2. Datasets

### A.2.1. Proteins

The data analysed as figures 3, 5 and 12 were derived from the European Protein Database TrEMBL v. 17-03 (March 2017) [48]. The data shown analysed as figure 15 were derived from versions 15-07, 17-03 and 19-04.

**Table 2.** The protein 20 kDa chaperonin, chloroplastic, from gene CPN21 of *Pisum sativum* (garden pea) https://www.uniprot.org/uniprot/P31233.

| protein |
| --- |
| ATVVAPKYTAIKPLGDRVLVK |

TrEMBL is a large and rapidly growing database which is updated monthly and the last release we used was 19-04 as shown in figure 15 which contained some 156 million proteins.

An example of a short protein is shown in table 2. Proteins appear as sequential strings of letters in a 22-letter alphabet, each letter corresponding to an amino acid encoded from the DNA of the organism; for example, the letter V corresponds to the amino acid valine. The theory we develop here does not depend in any way on the physico-chemical properties of the amino acids. They are treated simply as symbols with no other property than that they can be distinguished.

### A.2.2. Computer software

In the 50 or so years since they first appeared, many programming languages have arisen, from which computer programs of almost limitless functionality are built. We have already seen that individual *symbols* of a programming language are called *tokens*. They may take two forms: the *fixed* tokens of the language as provided by the language designers, and the *variable* tokens [49,50] which programmers supply as identifier and constant names. From our point of view, this distinction is irrelevant, they are all just tokens. There are many programming languages but all obey the same principles and every form of software system evolves from such tokens. Note that language tokens are syntactically indivisible, even though in programming languages they can comprise one or more characters.

Computer programs are often large. The software deployed in the search for the recently discovered Higg's boson comprises around 4 million lines of code [51]. At an average of around six tokens per line of code, this corresponds to some 24 million tokens, although this is still less than 1% of the human genome in which the tokens are the four bases adenine, cytosine, guanine and thymine. The largest systems in use today appear to be in the tens of millions of lines of source code [52], corresponding to perhaps 15% of the number of tokens of the human genome. The population of open systems used to test the model described here total almost 100 million lines (specifically 98 476 765 lines), containing some 600 million tokens.

As an example of the nomenclature, consider the following simple program written in the language C which, given an integer will return its incremented value:

```
int addOne(int i) { return i + 1;}
```

This algorithm contains 13 tokens in all based on 8 of the fixed tokens and 3 of the variable tokens of ISO C, so the size of the unique alphabet for this component is $8 + 3 = 11$, (**int** and **i** are both repeated). We note in passing that unique alphabets in programming functions are usually much larger than the known unique alphabets of proteins, further differentiating between the two systems. Note finally that extracting the tokens of programming languages to assemble these measures requires the development of language compiler front-end tools [53,54]. This is caused by the fact that tokens can contain one or more characters and some knowledge of the syntax of a language is necessary to parse them. Analysing proteins is considerably simpler from a programming point of view as each token is simply a letter.

### A.2.3. Music

Modern digital formats for musical annotation such as MusicXML (https://en.wikipedia.org/wiki/MusicXML (accessed 7 July 2017)), are yet another distinct discrete system where, in this case, the components are pieces of music and the unique alphabet comprises notes as shown in table 1, although we note that this representation is verbose compared with either that of proteins or even computer software.

Extracting the appropriate data is relatively simple—intermediate in difficulty between computer software and proteins—as the following XML snippet shows, taken from "Nun danket alle Gott",

(Words Rinkart 1636, Music Crüger, 1647). (https://hymnary.org/media/fetch/99378 (accessed 7 July 2017))

```
<part id="P1">
  <measure number="1">
  ...
    <note>
      <pitch>
        <step>E</step>
        <alter>-1</alter>
        <octave>4</octave>
      </pitch>
      <duration>480</duration>
      <voice>1</voice>
    ...
```

This fragment refers to the note Eb in the 4th octave (middle C is annotated C4, so this is a minor third (3 semitones) above middle C). The duration must be determined in conjunction with other parameters in the XML but this note actually corresponds to a 1/4 note or crotchet. MusicXML is verbose but mostly easy to parse even though note timing is determined by more than one set of parameters.

In this study, we used 883 pieces of music, mostly classical but a very eclectic mix of chorales, piano concertos, horn duets, blue-grass music and indeed almost anything in an XML format we could get our hands on, but this dataset is still very much smaller than the others at around $8 \times 10^5$ tokens. The scale-independence of the theory, however, means that this is sufficient.

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
