## [Reviewer comments · Royal Society Open Science]

Review History

RSOS-191101.R0 (Original submission)

Review form: Reviewer 1

Is the manuscript scientifically sound in its present form?

Yes

Are the interpretations and conclusions justified by the results?

Yes

Is the language acceptable?

Yes

Do you have any ethical concerns with this paper?

No

Have you any concerns about statistical analyses in this paper?

No

Recommendation?

Accept with minor revision (please list in comments)

Comments to the Author(s)

This is an interesting, well written paper that provides theoretical arguments and empirical evidence to support a novel hypothesis: discrete systems that conserve Hartley-Shannon information (called COHSI here) exhibit predictable distributions of component length, with specific predictions for heterogeneous systems and homogeneous systems, the latter reducing to Zipf's law. The theory is clearly presented, and I particularly appreciated the way in which Zipf's law appears as a special case. The paper argues convincingly that it has established the validity of the COHSI hypothesis. Yet at the same time it acknowledges that the link between the observed commonality of the length distributions and COHSI remains a hypothesis.

I enjoyed reading the paper and found it unusually insightful and accessible.
I strongly support publication.

I have several suggestions to make, all of which should be made at the authors' discretion:

1. To better communicate the ideas underlying COHSI, it would be helpful if the authors could briefly present a counter example that is not conservative.
2. The authors argue that there is no need for them to make comparisons with, e.g., lognormal or gamma distributions. I completely agree. However, in making such an argument, it would seem that the authors should be comparing their empirical distributions to the full theoretical distribution, not just the power-law tail. Why is this not done? I can imagine various difficulties, but I would appreciate an explicit discussion.
3. Along the lines of (2), it seems that the mode of the length distribution can be estimated from the properties of the specific discrete system being studied. Figure 9 implies this. I would appreciate some discussion of what insight, if any, may be inferred from the component length associated with the model.

Finally, the graphs should be more clear:

1. The labels in Figure 9 and the other figures is far too small. These labels should be increased in size following typical standards.
2. The purple region of Figure 9 should be explained in the caption. The legend contains 3 lines but it is not clear visually that three different curves are in the figure.

Review form: Reviewer 2

Is the manuscript scientifically sound in its present form?

Yes

Are the interpretations and conclusions justified by the results?

Yes

Is the language acceptable?

Yes

Do you have any ethical concerns with this paper?

No

Have you any concerns about statistical analyses in this paper?

No

Recommendation?

Accept as is

Comments to the Author(s)

This is an important manuscript and one that finds a very natural fit with the goals and format of RS Open Science publication. I don't have any hesitation in recommending it for publication in its present form. I have some considerations I would like to offer but, as I write them, I realize that these are the exact sort of questions and issues that the two authors of this proposal for a unifying information conservation principle for tokenized systems want to provoke. Most important, this is a proposal backed by extensive numeric experimentation with tokenized systems data - from Molecular Biology to Music and Programming languages - and with open source tools contributed by the authors to the public domain. I can't imagine a more fitting approach for dissemination by an open science publication.

The search for unifying theories, or even representations, of what exactly persists in a persistently interconnected systems has a long, deep, interdisciplinary history. In an era of widely available big data, any progress in that exploration has a sense of urgency. Progress, however, has been slow. This report starts by a pointed Introduction stating the facts - that scale-free power-law relationships are everywhere for one to see, and that early inroads into explaining them, such as Self-Organizing Criticality are increasingly "untenable" in explaining or unifying observations. This is a point that will be argued back by those using it to identify objective criteria for optimal systems organization, such as "Optimization by Self-Organized Criticality" by Hoffmann and Payton in Nature's Scientific Reports last year. This report leaves that argument for after the facts, and indeed after publication - this theoretical proposal ends up being very much about facts. The unavoidable fact is that across disciplines we see deviations from Zipf's law that the CoHSI tackles head-on as the criteria for what a modern theory for discrete systems must satisfy. They are absolutely right, as the data has become bigger, the evidence for tail deviations has made current interpretation correspondingly more untenable.

The CoHSI makes a second important contribution and one that derives directly from the experimentally observed discordant evidence at the tails of Zipf's log-log linearity. It identifies those discordances as reflecting two distinct system behaviors (which may, or may not be present in the same system): heterogeneous and homogeneous. The distinction relies on the nature of the tokenization of the components - clearly heterogeneous in Biological Systems such as protein data, for example, and clearly constrained in symbolic representations such as languages (music and computer code in the analysis). Simply put, this is an explanation that holds explainability in the numeric analysis that drives this proposal - what more could one ask from a theory! The long component tail in heterogeneous systems, as the authors recall, simply cannot be ignored by a unifying systems theory. Correspondingly, the mathematical analysis in section #3 leading to a mathematical description of information conservation in the two CoHSI distributions is compelling and its resolution in section #4 (Results) passes the litmus test of including those tail "abnormalities".

Sections #3 and #4 earn the authors a number of important claims in the Conclusions (section #5). Different readers will find different value in each of them - as it would be expected by the transdisciplinary nature of this study. I find the third, the categorization bias, particularly

striking. We tend to categorize systems as being heterogeneous or homogeneous with no clear method beyond the Cartesian argument that it is “obvious”. CoHSI proposes instead that we let the nature of the tokenization answer that question for us. It then becomes a distinction with a symptomatic tail distribution. That is a powerful contribution in an era where ever-increasing datasets, in volume and complexity, signifies that the analysis is performed by individuals and analytical infrastructure not versed or configured for the domain behaviors they seek to describe.

Decision letter (RSOS-191101.R0)

16-Sep-2019

Dear Dr Hatton

On behalf of the Editors, I am pleased to inform you that your Manuscript RSOS-191101 entitled "Strong evidence of an Information-Theoretical Conservation Principle linking all discrete systems" has been accepted for publication in Royal Society Open Science subject to minor revision in accordance with the referee suggestions. Please find the referees' comments at the end of this email.

The reviewers and handling editors have recommended publication, but also suggest some minor revisions to your manuscript. Therefore, I invite you to respond to the comments and revise your manuscript.

- Ethics statement

- Data accessibility

If you wish to submit your supporting data or code to Dryad (<http://datadryad.org/>), or modify your current submission to dryad, please use the following link:
<http://datadryad.org/submit?journalID=RSOS&manu=RSOS-191101>

- Competing interests

- Authors' contributions

All submissions, other than those with a single author, must include an Authors' Contributions section which individually lists the specific contribution of each author. The list of Authors

should meet all of the following criteria; 1) substantial contributions to conception and design, or acquisition of data, or analysis and interpretation of data; 2) drafting the article or revising it critically for important intellectual content; and 3) final approval of the version to be published.

- Acknowledgements

- Funding statement

Because the schedule for publication is very tight, it is a condition of publication that you submit the revised version of your manuscript before 25-Sep-2019. Please note that the revision deadline will expire at 00.00am on this date. If you do not think you will be able to meet this date please let me know immediately.

- 1) A text file of the manuscript (tex, txt, rtf, docx or doc), references, tables (including captions) and figure captions. Do not upload a PDF as your "Main Document";

- 2) A separate electronic file of each figure (EPS or print-quality PDF preferred (either format should be produced directly from original creation package), or original software format);
- 3) Included a 100 word media summary of your paper when requested at submission. Please ensure you have entered correct contact details (email, institution and telephone) in your user account;
- 4) Included the raw data to support the claims made in your paper. You can either include your data as electronic supplementary material or upload to a repository and include the relevant doi within your manuscript. Make sure it is clear in your data accessibility statement how the data can be accessed;
- 5) All supplementary materials accompanying an accepted article will be treated as in their final form. Note that the Royal Society will neither edit nor typeset supplementary material and it will be hosted as provided. Please ensure that the supplementary material includes the paper details where possible (authors, article title, journal name).

on behalf of Dr Francois Fages (Associate Editor) and Marta Kwiatkowska (Subject Editor)
openscience@royalsociety.org

Associate Editor Comments to Author (Dr Francois Fages):

Associate Editor: 1

Comments to the Author:

Dear authors

It is my pleasure to accept your paper with just minor revision.

Please take into account the attached reviews in producing your final version.

Best regards

Reviewer comments to Author:

Reviewer: 1

Comments to the Author(s)

This is an interesting, well written paper that provides theoretical arguments and empirical evidence to support a novel hypothesis: discrete systems that conserve Hartley-Shannon information (called COHSI here) exhibit predictable distributions of component length, with specific predictions for heterogeneous systems and homogeneous systems, the latter reducing to Zipf's law. The theory is clearly presented, and I particularly appreciated the way in which Zipf's law appears as a special case. The paper argues convincingly that it has established the validity of the COHSI hypothesis. Yet at the same time it acknowledges that the link between the observed commonality of the length distributions and COHSI remains a hypothesis.

I enjoyed reading the paper and found it unusually insightful and accessible.

I strongly support publication.

I have several suggestions to make, all of which should be made at the authors' discretion:

1. To better communicate the ideas underlying COHSI, it would be helpful if the authors could briefly present a counter example that is not conservative.
2. The authors argue that there is no need for them to make comparisons with, e.g., lognormal or gamma distributions. I completely agree. However, in making such an argument, it would seem that the authors should be comparing their empirical distributions to the full theoretical distribution, not just the power-law tail. Why is this not done? I can imagine various difficulties, but I would appreciate an explicit discussion.
3. Along the lines of (2), it seems that the mode of the length distribution can be estimated from the properties of the specific discrete system being studied. Figure 9 implies this. I would appreciate some discussion of what insight, if any, may be inferred from the component length associated with the model.

Finally, the graphs should be more clear:

1. The labels in Figure 9 and the other figures is far too small. These labels should be increased in size following typical standards.
2. The purple region of Figure 9 should be explained in the caption. The legend contains 3 lines but it is not clear visually that three different curves are in the figure.

Reviewer: 2

Comments to the Author(s)

This is an important manuscript and one that finds a very natural fit with the goals and format of RS Open Science publication. I don't have any hesitation in recommending it for publication in its present form. I have some considerations I would like to offer but, as I write them, I realize that these are the exact sort of questions and issues that the two authors of this proposal for a unifying information conservation principle for tokenized systems want to provoke. Most important, this is a proposal backed by extensive numeric experimentation with tokenized systems data - from Molecular Biology to Music and Programming languages - and with open source tools

contributed by the authors to the public domain. I can't imagine a more fitting approach for dissemination by an open science publication.

The search for unifying theories, or even representations, of what exactly persists in a persistently interconnected systems has a long, deep, interdisciplinary history. In an era of widely available big data, any progress in that exploration has a sense of urgency. Progress, however, has been slow. This report starts by a pointed Introduction stating the facts - that scale-free power-law relationships are everywhere for one to see, and that early inroads into explaining them, such as Self-Organizing Criticality are increasingly "untenable" in explaining or unifying observations. This is a point that will be argued back by those using it to identify objective criteria for optimal systems organization, such as "Optimization by Self-Organized Criticality" by Hoffmann and Payton in Nature's Scientific Reports last year. This report leaves that argument for after the facts, and indeed after publication - this theoretical proposal ends up being very much about facts. The unavoidable fact is that across disciplines we see deviations from Zipf's law that the CoHSI tackles head-on as the criteria for what a modern theory for discrete systems must satisfy. They are absolutely right, as the data has become bigger, the evidence for tail deviations has made current interpretation correspondingly more untenable.

The CoHSI makes a second important contribution and one that derives directly from the experimentally observed discordant evidence at the tails of Zipf's log-log linearity. It identifies those discordances as reflecting two distinct system behaviors (which may, or may not be present in the same system): heterogeneous and homogeneous. The distinction relies on the nature of the tokenization of the components - clearly heterogeneous in Biological Systems such as protein data, for example, and clearly constrained in symbolic representations such as languages (music and computer code in the analysis). Simply put, this is an explanation that holds explainability in the numeric analysis that drives this proposal - what more could one ask from a theory! The long component tail in heterogeneous systems, as the authors recall, simply cannot be ignored by a unifying systems theory. Correspondingly, the mathematical analysis in section #3 leading to a mathematical description of information conservation in the two CoHSI distributions is compelling and its resolution in section #4 (Results) passes the litmus test of including those tail "abnormalities".

Sections #3 and #4 earn the authors a number of important claims in the Conclusions (section #5). Different readers will find different value in each of them - as it would be expected by the transdisciplinary nature of this study. I find the third, the categorization bias, particularly striking. We tend to categorize systems as being heterogeneous or homogeneous with no clear method beyond the Cartesian argument that it is "obvious". CoHSI proposes instead that we let the nature of the tokenization answer that question for us. It then becomes a distinction with a symptomatic tail distribution. That is a powerful contribution in an era where ever-increasing datasets, in volume and complexity, signifies that the analysis is performed by individuals and analytical infrastructure not versed or configured for the domain behaviors they seek to describe.

Author's Response to Decision Letter for (RSOS-191101.R0)

See Appendix A.

Decision letter (RSOS-191101.R1)

24-Sep-2019

Dear Dr Hatton,

I am pleased to inform you that your manuscript entitled "Strong evidence of an Information-Theoretical Conservation Principle linking all discrete systems" is now accepted for publication in Royal Society Open Science.

on behalf of Dr Francois Fages (Associate Editor) and Marta Kwiatkowska (Subject Editor)
openscience@royalsociety.org

Follow Royal Society Publishing on Twitter: [@RSocPublishing](https://twitter.com/RSocPublishing)

Appendix A

Notes for Reviewers:

First of all, thank you for your valuable feedback on this MS. We hope we have satisfactorily answered the points you raised. We have included with the materials a pdf in which all the changes we made (a number of minor wording changes to clarify the arguments along with a couple of typos) have been highlighted in yellow. The points you raised are further discussed below.

Reviewer 1/1. To better communicate the ideas underlying COHSI, it would be helpful if the authors could briefly present a counter example that is not conservative.

This is certainly thought-provoking. We have yet to identify a naturally occurring dataset of substantial size which breaks CoHSI but they are easy to construct. We have therefore added an example in the Conclusions (at line 745) by extending the existing discussion that CoHSI is not a straitjacket. The comments follow:-

"It is perfectly possible within our theory for a discrete system to have a different length distribution other than that predicted by CoHSI; CoHSI only identifies the overwhelmingly most likely distribution within its constraints. This can be seen clearly by construction in the domain of computer software. In a matter of seconds, the simple computer program shown in the Appendix could be copied several million times and the result deemed a system in our nomenclature. Artificial though it may seem, this is a perfectly valid system in a theory which is agnostic with regard to the tokens - the functionality is irrelevant. This system manifestly does not satisfy CoHSI, but is simply unlikely in the ensemble of all systems with the same number of tokens T and the same information content I . In the natural world too, natural selection could force a departure from the most likely distribution implied by CoHSI, which we interpret as CoHSI acting only to guide and not to enforce."

Reviewer 1/2. The authors argue that there is no need for them to make comparisons with, e.g., lognormal or gamma distributions. I completely agree. However, in making such an argument, it would seem that the authors should be comparing their empirical distributions to the full theoretical distribution, not just the power-law tail. Why is this not done? I can imagine various difficulties, but I would appreciate an explicit discussion.

Reviewer 1/3. Along the lines of (2), it seems that the mode of the length distribution can be estimated from the properties of the specific discrete system being studied. Figure 9 implies this. I would appreciate some discussion of what insight, if any, may be inferred from the component length associated with the model.

Thank you for bringing this omission to our attention. There are indeed difficulties and this is an ongoing area of research. We have added some comments in the discussion around Fig. 9 (starting at line 478) to address this as follows:

"This solution is explored for different values of the Lagrange parameters in \cite{HattonWarr2018a}. In essence, both Lagrange parameters contribute to the shape and position of the unimodal peak because of the implicit nature of the solutions and there indeed may be no solutions for the smallest values of σ_{i} for some values. This is certainly the case with the datasets we used - for example a total of only 6 sequences shorter than 7 amino acids is included in the TrEMBL 19-04 database and the smallest legal component in the programming language C for example contains 3 tokens, but it exists only as an oddity as it has no functional behaviour. Whilst fitting the asymptotic power-law is relatively simple, the implicit nature of (\ref{eq:cohsi}) and other complications such as the scale of the numerics and the subtle relationship between the Lagrangian parameters α, β , the alphabet of tokens and corresponding parameters in the real data as revealed in \cite{HattonWarr2018a}, present significant challenges for quantitative fitting of our real datasets to the shape and position of the mode in the CoHSI heterogeneous equation and this remains a work in progress."

We hope this gives a satisfactory explanation at our current state of understanding.

Finally, the graphs should be more clear:

1. The labels in Figure 9 and the other figures is far too small. These labels should be increased in size following typical standards.
2. The purple region of Figure 9 should be explained in the caption. The legend contains 3 lines but it is not clear visually that three different curves are in the figure.

These have been attended to as suggested and we have standardised on a Helvetica 24 font for axis annotation and a Helvetica 18 font for the key and also the axis scale annotations. The purple region of Figure 9 is now explained in the figure caption as suggested and the wording has been changed a little to make the presence of 3 curves more clear.

Reviewer 2 did not make specific points but contained a number of very useful insights that made us realise that some parts of our argument would benefit from a little more emphasis. One or two of the highlighted changes, notably in the final point of the Conclusions reflect these insights.